# The Importance of Wake Meandering on Wind Turbine Fatigue Loads in Wake

Jennifer Marie Rinker [1,*,†] ![ID], Esperanza Soto Sagredo [1,†] and Leonardo Bergami [2] ![ID]

1 Department of Wind Energy, Technical University of Denmark, Anker Engelunds Vej 1 Bygning 101A, 2800 Kongens Lyngby, Denmark; esotosagredo@gmail.com
2 Conceptual Design, LM Wind Power, Jupitervej, 6000 Kolding, Denmark; Leonardo.Bergami@lmwindpower.com
* Correspondence: rink@dtu.dk
† These authors contributed equally to this work.

**Abstract:** Considering loads when optimizing wind-farm layouts or designing farm-control strategies is important, but the computational cost of using high-fidelity wake models in the loop can be prohibitively high. Using simpler models that consider only the spatial variation of turbulence statistics is a tempting alternative, but the accuracy of these models with respect to the aeroelastic response is not well understood. This paper therefore highlights the effect of replacing wake meandering with spatially varying statistics ("profile functions") in the inflow to a downstream turbine. Profile functions at different downstream and lateral locations are extracted from a large-eddy simulation with an upstream turbine and compared with two lower-fidelity models: one that prescribes both the mean and standard deviation of the turbulence and one that prescribes only the mean. The aeroelastic response of an NREL 5 MW wind turbine is simulated with the three different wake-model fidelities, and various quantities of interest are compared. The mean values for the power and rotor speed for the medium-and low-fidelity model match well, but the accuracy of the fatigue loads varies greatly depending on the load channel. Prescribing the profile function for the standard deviation is only beneficial for the tower-base fore-aft moment; all other DELs had similar accuracies for both the medium- and low-fidelity models. The paper concludes that blade DELs can be estimated using these simple models with some accuracy, but care should be taken with the load channels related to the shaft torsion and tower-base fore-aft bending moment.

**Keywords:** wind turbine loads; wind turbine wakes; turbulence





## 1. Introduction

The main driver in the development of wind turbine technology is the reduction in the cost of wind energy, whether measured in terms of the levelized cost of energy (LCOE), internal rate of return (IRR) or some other economic metric [1]. As more and more turbines are placed in closely clustered wind farms, this emphasis on cost reduction has expanded from a single-turbine view to a farm-level view. For wind farms already erected and operating, reductions in cost could be achieved via recent developments in farm-level control. Many recent publications that have investigated wind farm control have concluded that various techniques can significantly increase wind farm production [2–4], and industry has also begun to implement farm-level control techniques such as wake steering in their installations [5]. For wind plants that are currently being designed, novel layout optimization techniques provide further opportunities to increase plant production [6,7].

Although there is a plethora of literature focusing on techniques to improve plant production, the vast majority of papers do not consider an essential aspect: wind turbine loads. In the area of wind-farm control, only a selection of papers have considered the impact of the proposed plant control strategy on loads [8–13]. Of those papers, only three consider the

entire plant [8–10], while the other papers consider only a single upstream turbine [11–13]. There is a similar dearth of research literature for loads-aware plant-layout optimization: only two papers have considered loads during plant-layout optimization [14,15]. Of those two papers, one presented only a single, simple optimization layout [14] and the other considered the fatigue loads for only one load channel: the blade-root edgewise moment [15]. The lack of robust publications in this area makes it difficult to draw concrete conclusions about whether the potential increase in plant production is negated by a correlated increase of wind turbine loads.

The general lack of the body of research is not the only issue hindering wind energy development: except for [8,9], all of the publications mentioned above use steady-state wake models for their investigations. The use of steady-state wake models is understandable in these applications, as using a higher-fidelity model such as large-eddy simulation (LES) or dynamic wake meandering [16] is prohibitively computationally expensive. Regardless, it has not been established in the literature whether one can use steady-state wake models to accurately calculate fatigue loads in waked turbines in farms. Thus, the objective of this paper is to determine the accuracy of a turbine's aeroelastic response when the inflow is calculated with LES versus lower-fidelity, steady-state wake models.

The remainder of this paper is organized as follows. First, the background for the LES data and the method for generating the turbulence boxes for different model fidelities is given in Section 2. Details on the aeroelastic modeling and wind turbine model are also provided in the section. The results are presented and analyzed in Section 3. The results include investigations into the importance of random seeds, the accuracy of the mean power/rotor speed for the different wake-model fidelities, and finally, the accuracy of the fatigue loads for the different wake-model fidelities. Section 4 discusses the most relevant aspects of the work as well as potential areas for further investigations, and the final conclusions are drawn in Section 5. More details on some of the results presented in this paper can be found in [17].

## 2. Methodology

The impact of wake meandering on the aeroelastic response of downstream turbines is investigated in this paper by running aeroelastic simulations using inflow with wake effects modeled using three different levels of fidelity. The turbulence boxes with the highest fidelity were extracted directly from LES data and therefore included dynamic wake effects such as meandering. The turbulence boxes with the two lower-fidelity models only included steady wake effects and were created using PyConTurb [18] (more details in Section 2.3). Because the wake effects change based on the location of the downstream turbine, the lateral location and downstream distance of the downwind turbine were varied to investigate the impacts of partial-wake scenarios at different downstream distances.

The details of the methodology are presented in the following subsections. First, the LES data from which the high-fidelity turbulence boxes were extracted is described in Section 2.1. The different lateral translations of the downwind turbine—which allowed the investigation of partial- and full-wake scenarios—are defined in Section 2.2. Section 2.3 describes the three different model fidelities used to generate inflow with wake characteristics. Lastly, the details of the aeroelastic simulations and post-processing are given in Section 2.4.

### 2.1. LES Data

The high-fidelity turbulence boxes used in this study are turbulence boxes extracted directly from LES data and formatted to be compatible with HAWC2 (Horizontal Axis Wind turbine simulation Code 2nd generation, [19]). The National Renewable Energy Laboratory (NREL) simulated the LES results using their in-house computational fluid dynamics (CFD) code SOWFA (Simulator fOr Wind Farm Applications [20]), extracted the HAWC2-style turbulence boxes and provided the boxes to the authors. All SOWFA simulations used in this paper had a mean wind speed of 8 m/s and included a single

NREL 5MW Reference Wind Turbine [21] that was operating with a fixed pitch angle of 0 degrees and a fixed rotational speed of 9.1 rpm. Two separate turbulence intensities were simulated in SOWFA by modifying the surface roughness: a low-TI case with an average TI of 5.6% and a high-TI case with an average TI of 11.0%. The parameters used in the SOWFA simulations are given in Appendix A.

The HAWC2 turbulence boxes were extracted at four different downwind distances: m2*D* (-2*D*), 4*D*, 7*D* and 10*D*, where *D* indicates the rotor diameter. The m2*D* position is two rotor diameters upwind of the NREL 5MW and is assumed to represent the freestream case. The authors acknowledge that 2*D* upstream may not be a sufficient distance from the turbine to be considered fully freestream. However, because these slices are only used to normalize the downstream results, this assumption is acceptable. The other three cases (4*D*, 7*D* and 10*D*) are different distances downwind, which allows the investigation of wake effects produced by the upstream turbine for different downwind distances. The high-fidelity HAWC2 turbulence boxes are 400 m wide by 300 m tall with a spacing of 2.5 m (161 by 121 points). The time series have 28,000 points with a $\Delta t$ of 0.02 s for a total simulation time of 700 s. A diagram of the turbulence box and related coordinate systems is given in Figure 1. Because there are two turbulence intensities and four along-wind locations, there are eight high-fidelity HAWC2 boxes.

### 2.2. Lateral Location of Downstream Turbine

One of the main research questions investigated in this paper is how the partial- and full-wake scenarios impact operational data and loads. To investigate the answer to this question, the aeroelastic response of the downstream turbine was simulated at seven different lateral locations: $Y = -126$ m, $-63$ m, $-30$ m, 0 m, 30 m, 63 m and 126 m, where $Y$ is defined as shown in Figure 1. An example of the YZ plane with different lateral turbine locations is given in Figure 2. The position of the upstream turbine is indicated by the gray dashed circle in Subplot (a), and the colored contours indicate the mean wind speed for the low-TI LES data at 2*D* upstream (a) or 4*D* downstream (b through h). Combining the eight high-fidelity boxes and the seven lateral turbine locations yields 56 different combinations of high-fidelity aeroelastic simulations.

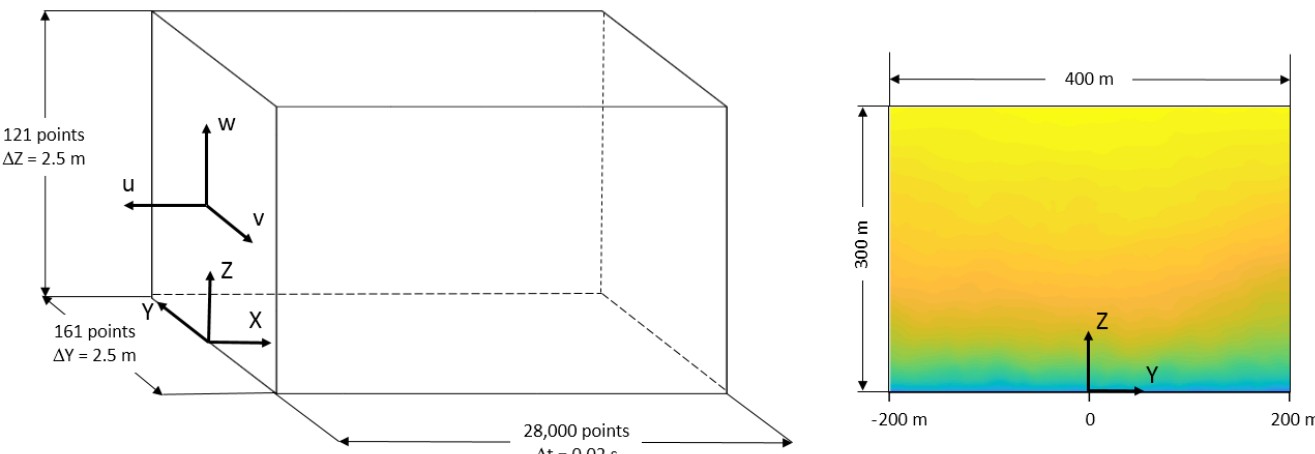

**Figure 1.** Turbulence box size and coordinate systems used in this study (**left**) and an example mean-wind-speed profile when looking downwind (**right**). The $u$, $v$, $w$ coordinate system corresponds to the standard turbulence components, consistent with PyConTurb.

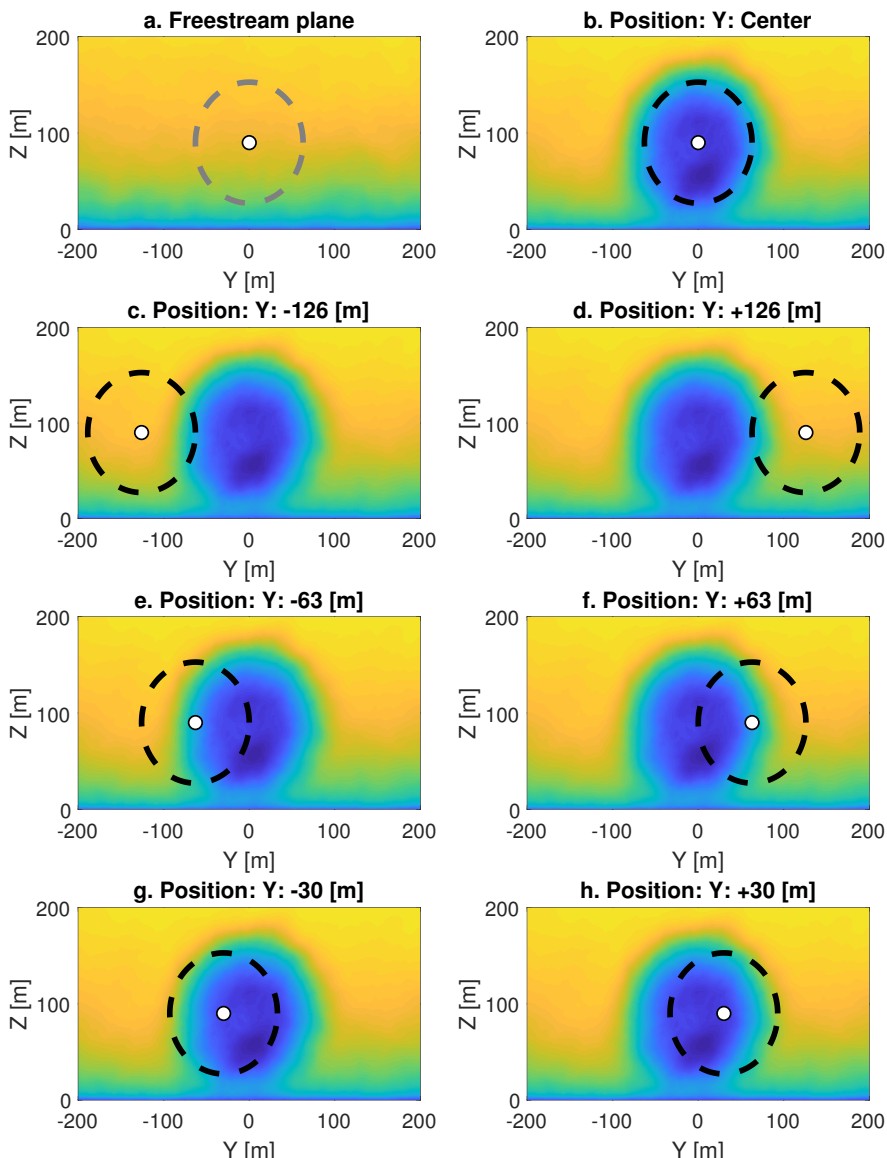

**Figure 2.** Selection of downstream turbine positions in the YZ plane for different wake cases (e.g., partial- or full-wake). Dashed lines indicate turbine positions, and contours indicate mean wind speed for low-TI case either 2*D* upstream (**a**) or 4*D* downstream (**b**–**h**).

### 2.3. Fidelity of Different Turbulence Boxes

As noted above, the high-fidelity boxes were extracted from the LES data, whereas the medium- and low-fidelity boxes were generated using PyConTurb, an open-source software for creating constrained turbulence boxes [18]. All turbulence boxes used in the aeroelastic simulation were 126 m wide and 126 m tall.

There are two general methods in PyConTurb for constraining turbulence boxes to measurements: by using "profile functions" or direct time-series constraints. Profile functions are functions that prescribe the variation of the mean wind speed, turbulence standard deviation and/or power spectrum as a function of lateral and vertical locations in the rotor plane. Time-series constraints use the method described in [22] to spatially correlate the Fourier components of the new simulation points to those already existing in the constraints. Although using time-series constraints is more accurate than profile

functions and can reproduce interesting transient phenomena, this paper focuses on steady-state wake models and therefore utilizes profile functions on the mean wind speed and turbulence standard deviation. The option to use profile functions on the power spectra is outside the scope of this paper.

The three different wake-modeling fidelities used in this paper are defined as follows:

- High fidelity. The high-fidelity turbulence boxes were directly extracted from the LES data as shown in Figure 3. Because they are taken from LES data, these turbulence boxes feature dynamic effects such as wake meandering. The power spectrum, standard deviation and mean wind speed are not specified directly, but rather are determined from the solution of the LES governing equations.
- Medium fidelity. The medium-fidelity boxes used profile functions for the mean wind speed and turbulence standard deviation calculated from the LES data. In other words, $U_{mf}(y,z) = U_{LES}(y,z)$ and $\sigma_{k,mf}(y,z) = \sigma_{k,LES}(y,z)$, where $y$ and $z$ are the lateral and vertical planar positions, respectively, and $k$ indicates the turbulence component (e.g., $u$, $v$ and $w$). Thus, the simulated turbulence box has the same spatial distribution in the YZ plane of the mean wind speed and a similar distribution of the standard deviation as the LES data. The standard deviation profile function is used to change the Fourier magnitudes before spatial correlation, but applying spatial correlation will unavoidably affect the standard deviation. This is one drawback of this constraint methodology. The temporal variations (i.e., along the X-direction of the box) created by PyConTurb assume stationary turbulence, and therefore dynamic effects such as wake meandering will be lost.
- Low fidelity. The low-fidelity boxes were constrained using only the profile functions for the mean wind speed extracted from the LES data; i.e., $U_{lf}(y,z) = U_{LES}(y,z)$ for all locations in the YZ plane. The standard deviation is no longer spatially varying, but rather set at a constant value equal to the mean of the LES standard deviations: $\sigma_{k,lf}(y,z) = \sigma_{k,lf} = \overline{\sigma_{k,LES}(y,z)}$, where $k$ indicates the turbulence component.

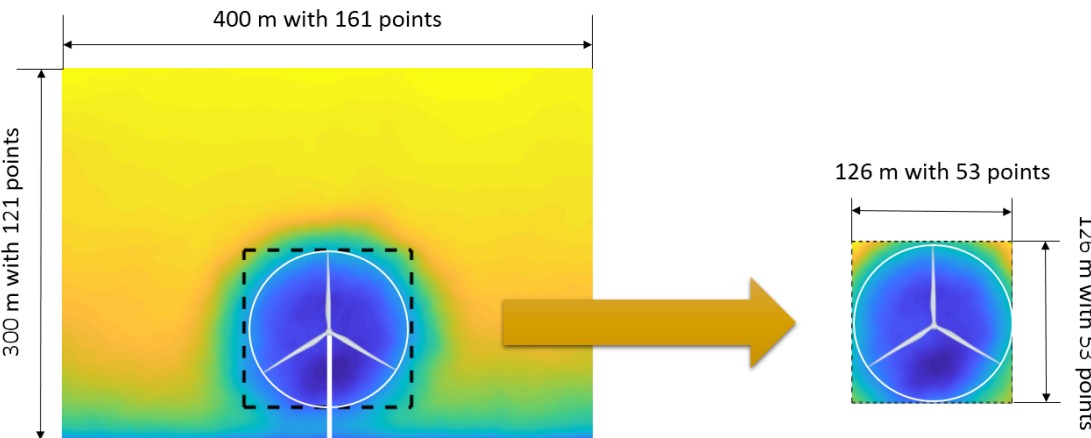

**Figure 3.** Extraction of high-fidelity turbulence box from LES data.

The low- and medium-fidelity turbulence boxes created in PyConTurb had $9 \times 9$ grid points in the YZ direction, which was chosen to balance computational accuracy and loads. A grid resolution study, not shown here for brevity (see [17]), revealed that the chosen grid resolution did not significantly impact the quantities of interest in this study. The prescribed spatial coherence model and power spectrum were the exponential coherence model and the Kaimal spectrum as specified in IEC 61400-1.

Generating the medium- and low-fidelity boxes required specifying a random seed for the random-number generation. To remove the influence of this random seed on the results, six medium-fidelity and low-fidelity boxes were generated with different random seeds for each high-fidelity box. Considering the original 56 high-fidelity boxes, this resulted in

336 turbulence boxes for each of the lower-fidelity models. Thus, there were a total of 728 10-min aeroelastic simulations for all three wake-model fidelities.

## 2.4. Aeroelastic Simulations and Fatigue Loads

The response of the turbine was simulated using version 12.6 of the aeroelastic software HAWC2 and the NREL 5 MW model provided on the HAWC2 website (www.hawc2.dk, accessed on 8 June 2021). The model was simulated for 700 s, from which the first 100 s were ignored to remove transience, using a time step of 0.02 s, which is the standard time step for this model. The turbulent inflow for each aeroelastic simulation was one of the 728 turbulence boxes, generated using the methodology described above.

The aeroelastic quantities of interest were the rotational speed and generated power, as well as the damage-equivalent fatigue loads (DELs) for the flapwise blade root moment, edgewise blade root moment, shaft torsion and tower-bottom fore-aft bending moment. A DEL represents the load amplitude that would have caused the same amount of damage for a reference number of cycles [23]. To determine the DEL from a load response, the response must be first converted to cycles and load amplitudes using the rainflow counting algorithm [24], after which the DEL for that time series can be calculated as follows:

$$M_{eq,i} = \left( \frac{\sum_{i=1}^{n_b} n_i M_i^m}{N} \right)^{\frac{1}{m}} \tag{1}$$

where $n_b$ is the number of bins used for the cycle counting, $n_i$ and $M_i$ are the number of cycles and load amplitude for bin $i$, respectively, $m$ is the material-dependent Wöhler exponent, and $N$ is the predefined number of reference cycles, which is assumed to be $10^6$ in this paper. The assumed Wöhler exponents were 4 for metal components (shaft, tower) and 10 for composite components (blades). The DELs for the six different seeds were combined to a single representative value by taking the mean over the six values. Once calculated for all wake-modeling fidelities and downstream locations, the DELs can be compared to determine how well the medium- and low-fidelity wake modeling methods reproduced the high-fidelity DELs at different turbine locations and for different channels.

## 3. Results

The results presented in this section are grouped into three subsections. First, it is of interest to quantify the variability present in the medium- and low-fidelity turbulence boxes caused by the choice of the random seed. To that end, the effect of the random seed is investigated in Section 3.1. Second, to yield more insight into the operational behavior of the turbine, the rotational speed and power generated in the aeroelastic simulations are compared in Section 3.2. Finally, the fatigue loads for the different wake-modeling fidelities and different turbine locations are compared in Section 3.3.

### 3.1. Influence of Random Seeds

As noted in Section 2.3, six random seeds were used to generate six different medium- and low-fidelity boxes for each downstream location, lateral location, and turbulence intensity. The same six seeds were used in all cases to allow a better comparison between the low- and medium-fidelity models.

The impact of the turbulence seed on the spatial distribution of the TI is shown in Figure 4. The box-plots indicate the variation of the TI over the YZ-plane, with the mean and median values indicated by the circles and horizontal lines, respectively. The rows correspond to different fidelities, the columns to different TIs in the LES simulation, and the "true" variation from the LES data is given by the lightest color. The mean and spread of the TI values decrease with increasing downstream distance, which matches intuition as the wake deficit decreases with increased distance. The impact of the random seed is clearly seen by the simulations with seed 97, which result in a significantly larger variation of TI over the rotor plane as well as a shifted mean value. The distribution of TI values

are generally significantly better with the medium-fidelity models, which is expected as the low-fidelity model does not prescribe the spatial variation of the standard deviation. This investigation not only yields insight into the accuracy of the two fidelities, but also highlights the importance of simulating with multiple seeds and combining the DELs from the different simulation, as is done in this paper.

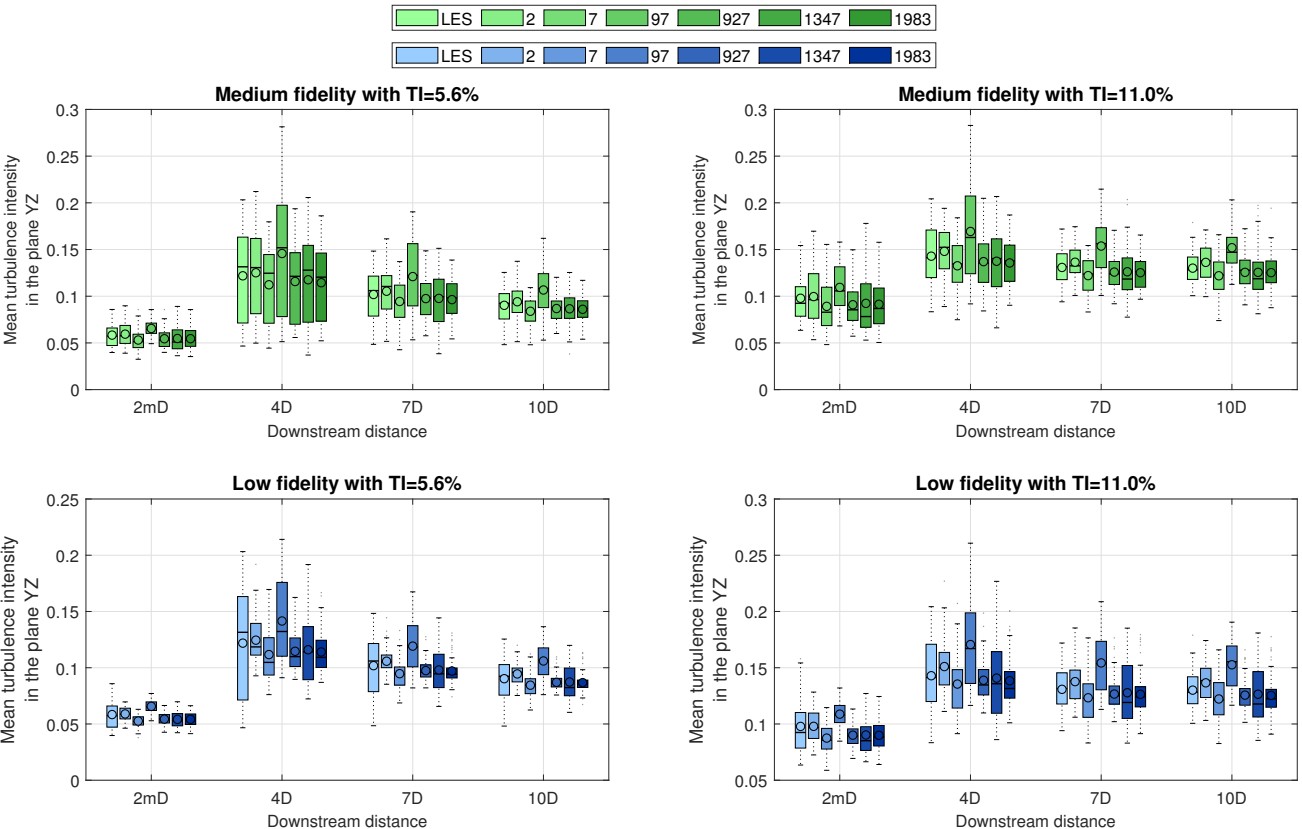

**Figure 4.** Influence of the random seed on the spatial distribution of the longitudinal turbulence intensity in a half-wake scenario. Box-plots indicate the variation of the TI in the YZ plane, circles indicate the mean TI, and the horizontal lines indicate the median TI. The top and bottom rows correspond to the medium- and low-fidelity models, respectively, and the left and right columns correspond to the low- and high-TI cases, respectively. The lightest color indicates the "true" spatial variation of the TI from the LES data.

### 3.2. Rotor-Equivalent Wind Speed, Power, and Rotor Speed

It is of interest to gain insight into the operational characteristics of the turbine before analyzing the fatigue loads with the different model fidelities. The rotor-equivalent wind speed (REWS) reflects the amount of available kinetic energy and is calculated according to [25,26]

$$U_{eq} = \sqrt[3]{\frac{\sum_j dA_j \cdot U_j^3}{A_{total}}},\qquad(2)$$

where $j$ is a horizontal slice of the rotor, and $A$ represents an area. The rotor-equivalent turbulence intensity (RETI) reflects the amount of turbulence across the rotor and is calculated by

$$I_{eq} = \sqrt{\frac{\sum_j dA_j \cdot I_j^2}{A_{total}}},\qquad(3)$$

where $I$ represents turbulence intensity.

The REWS and RETI values calculated from the high-fidelity turbulence boxes are visualized in Figure 5 for different downstream and lateral distances. There are two important characteristics of the freestream REWS for this LES simulation that are worth

noting: (1) for low TI, there is a very slight decrease in the freestream REWS as the turbine translates to the right, and (2) for high-TI, there is a parabolic-like shape to the freestream velocity. This spatial nonstationarity of the freestream LES data is due to the 700 s of the turbulence box being insufficient to achieve stationarity, and it is visual to the naked eye in the contour shown in Subplot (a) of Figure 2. The effect of the wake is seen as expected in both the REWS and RETI: in the wake situations, the REWS decreases and the RETI increases. The wake effects are generally less pronounced downstream due to wake recovery.

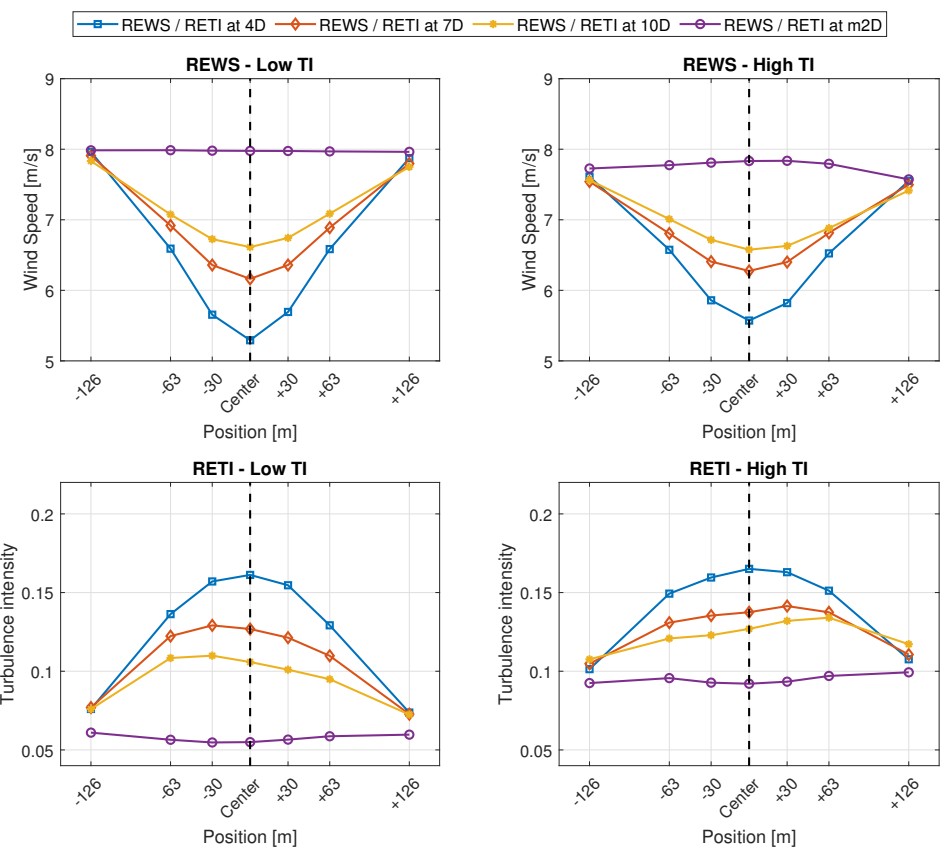

**Figure 5.** Rotor-equivalent wind speed (REWS; **top row**) and rotor-equivalent turbulence intensity (RETI; **bottom row**) for low-TI (**left**) and high-TI (**right**) cases. Values calculated from the high-fidelity boxes extracted from LES data.

The average power and rotational speed for the low-TI case and the three model fidelities is plotted in Figure 6. The black dashed line represents the simulations using the high-fidelity freestream case, and its downward slope reflects the previously noted trend in the freestream REWS trend for the low-TI case. Similar results for the high-TI case are shown in Figure 7, where the parabolic shape for the black dashed line reflects the parabolic shape of the high-TI REWS in Figure 5. A selection of the percent errors between the mean power from the high-fidelity simulations and the mean powers of the medium- and low-fidelity simulations is given in Table 1.

**Table 1.** Percent errors in the mean power with respect to the high-fidelity simulation for a selection of lateral and downstream locations.

| Location | Medium-Fidelity | Low-Fidelity |
|---|---|---|
| $[-63\text{ m}, 4D]$ | 7.67% | 8.31% |
| $[+63\text{ m}, 4D]$ | 5.87% | 6.35% |
| $[0\text{ m}, 4D]$ | 1.21% | 0.41% |
| $[-63\text{ m}, 7D]$ | 3.50% | 3.88% |
| $[-63\text{ m}, 10D]$ | 0.15% | 0.35% |

Several interesting observations can be made from the figures and table:

- The trends for both power and rotor speed generally follow the REWS trends shown in Figure 5 because the turbine is operating below rated.
- The medium- and low-fidelity models match each other quite closely, which is expected, as they have the same mean wind speed profile in the YZ-plane.
- The medium- and low-fidelity models match the high-fidelity model most closely in the full-wake scenarios. For the low-TI case, their accuracy increases with downstream distance. This is logical, as the medium- and low-fidelity models do not feature wake meandering, so lateral positions that are either primarily full-wake or no-wake will be more accurate than those with partial-wake scenarios.
- The turbine generates slightly more power at the $[+126, 4D]$ location than at freestream, a trend that is not visible in the REWS plot. The reason for this slight increase in power is likely due to the increased turbulence in the wake. The effect is reduced for larger downstream distances, as the REWS drops due to the larger meandering.

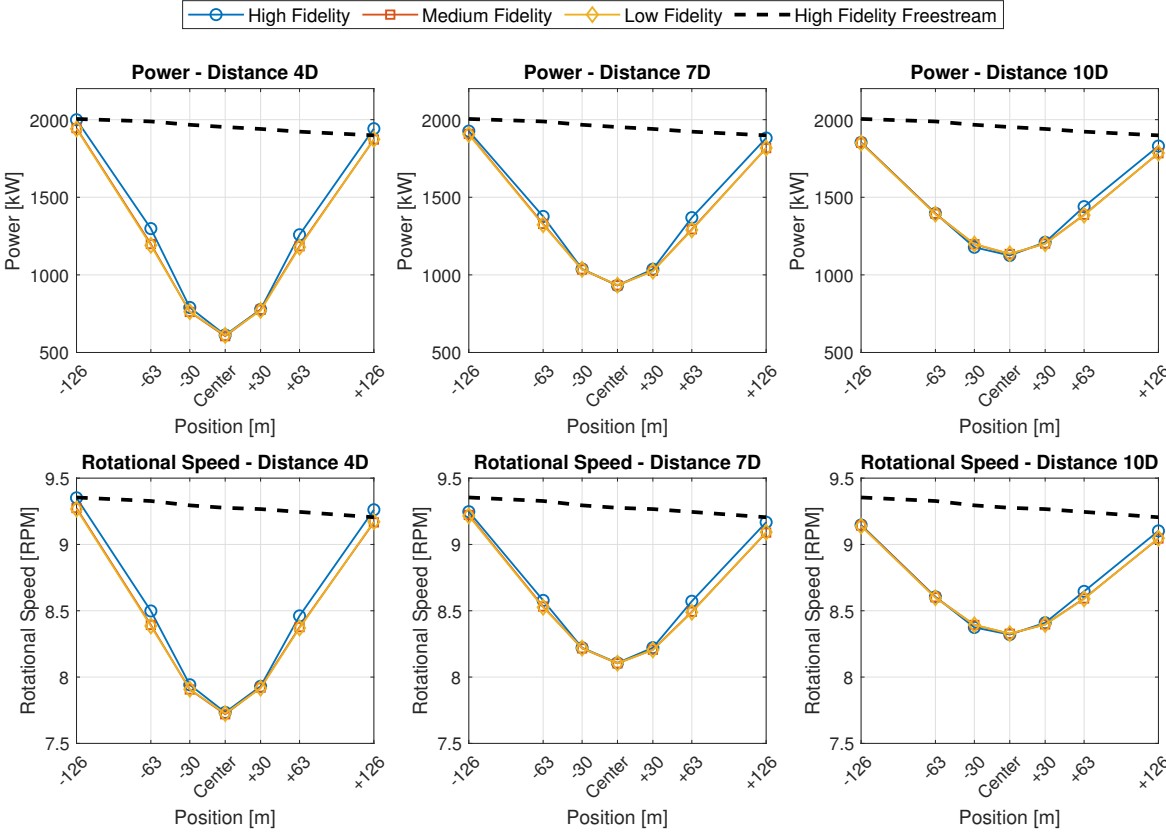

**Figure 6.** Mean power and rotor speed for the low-TI case. The dashed black line represents the high-fidelity freestream case (i.e., $2D$ in front of the rotor).

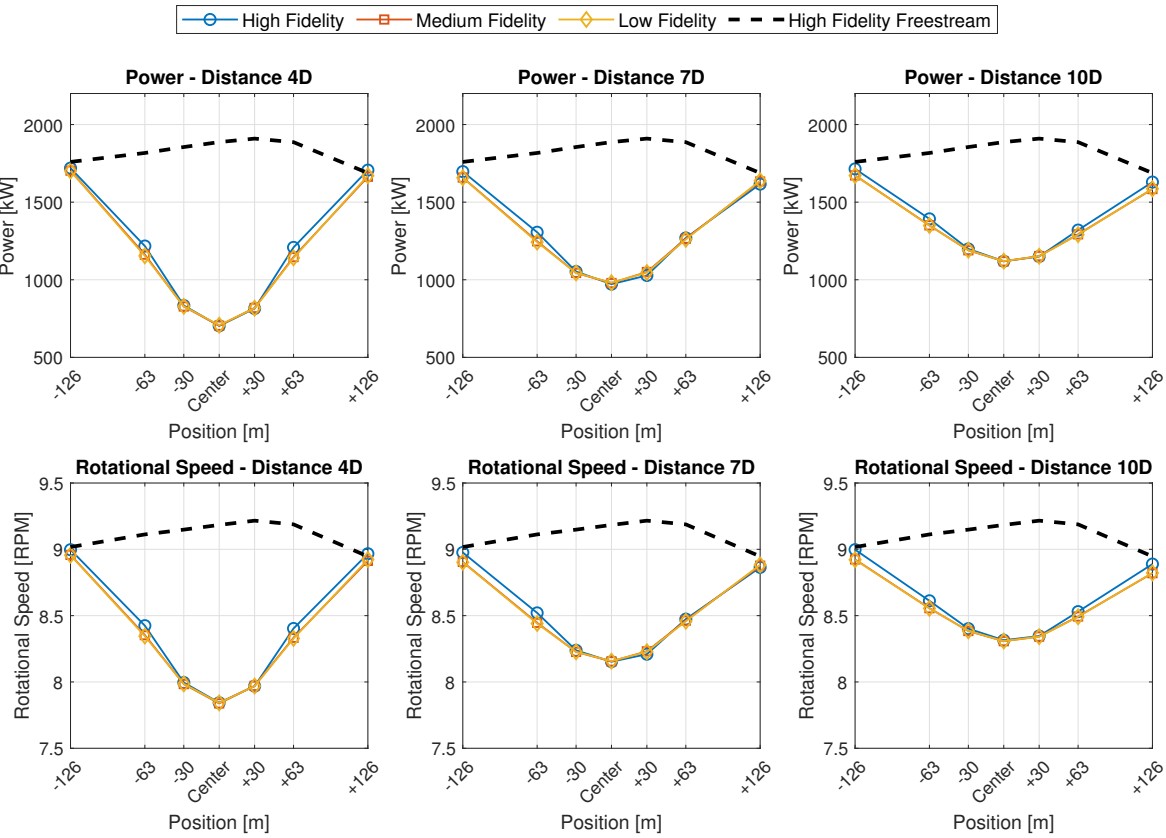

**Figure 7.** Mean power and rotor speed for the high-TI case. The dashed black line represents the high-fidelity freestream case (i.e., 2*D* in front of the rotor).

In summary, both the medium- and low-fidelity models are able to capture the general trends in the mean power and rotor speed of a turbine with a maximum 10% margin of error. The largest discrepancies occur in partial-wake scenarios, but the error drops to as low as 0.15% for full-wake scenarios and large downstream distances.

### 3.3. Fatigue Loads

The final step in the analysis is to evaluate how well the medium- and low-fidelity steady-state wake models can reproduce the fatigue loads of the LES high-fidelity model with meandering. For this analysis, representative fatigue DELs for each TI case and fidelity model are calculated according to the procedure explained in Section 2.4. To better facilitate a comparison across different model fidelities and downstream distances, all DELs are normalized by their respective high-fidelity, freestream DEL at the center location. The channels of interest are the blade-root edgewise and flapwise moments, the shaft torsion, and the tower-bottom fore-aft moment. Each channel of interest is discussed in a dedicated subsection below.

### 3.3.1. Blade-Root Flapwise Moment

The normalized DELs for the blade-root flapside moment are given in Figure 8. The results from the high-fidelity model (blue lines) reveal several interesting characteristics about these DELs. First, because the low-TI freestream DELs are significantly lower than the high-TI freestream DELs, we see much higher normalized DELs with the low-TI case than for the high-TI case. For example, a turbine placed at the −63 m transverse location with low TI would have flapwise DELs that were almost 6 times larger than a turbine in the freestream. Second, the DELs manifest in an M-like shape, especially for smaller downstream distances. The large DELs at the left and right peaks are due to the partial-wake scenarios, which significantly increase periodic loading on the blade and thereby increase

the DELs. The left-right asymmetry in the "M" shape is especially noticeable in the low-TI case and is caused by interactions between the shear, blade rotational direction, and wake. The spreading of the wake at farther downstream distances reduces the amplitude of the M-shape, as the wake location varies to larger lateral positions in the plane. Finally, it is interesting to note that the high-TI DELs near the center are actually lower downstream than in the freestream, as their normalized DELs are less than unity. This is due to the wake deficit, which decreases the wind shear contribution to fatigue, thereby counteracting the increased turbulence intensity in the wake compared to the freestream.

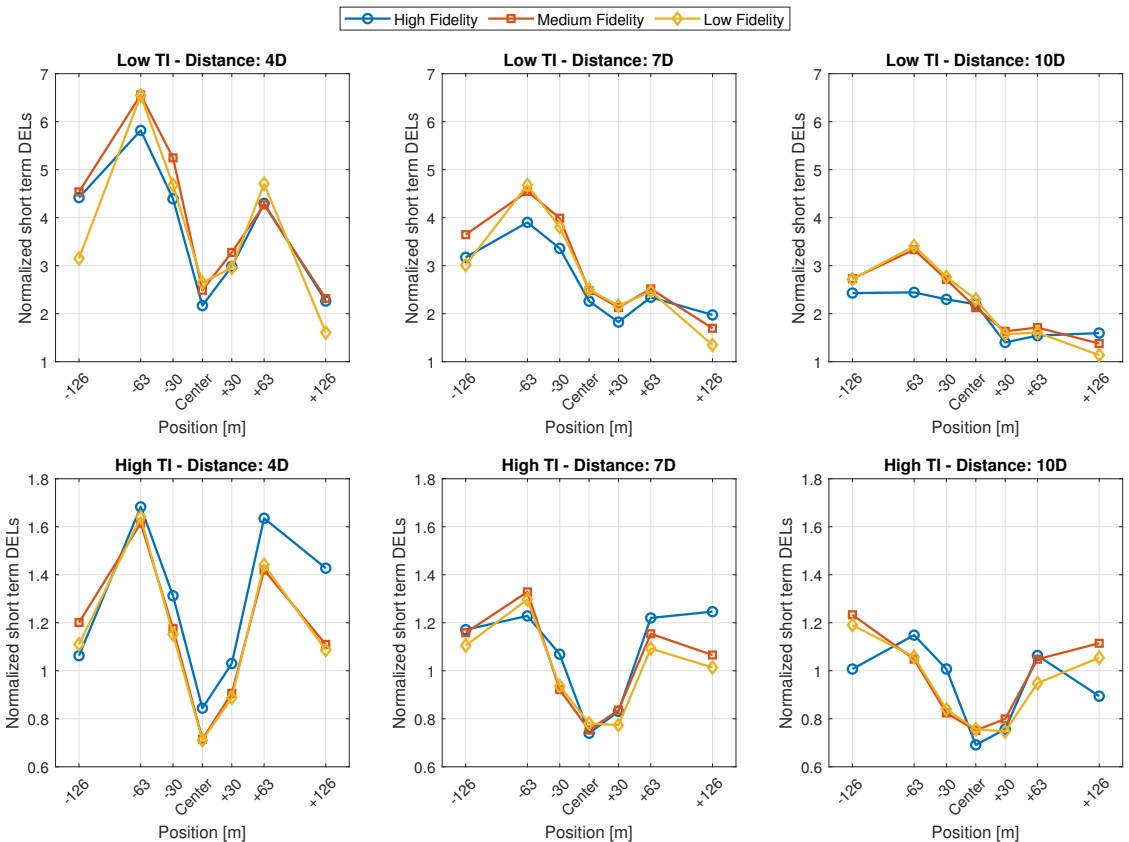

**Figure 8.** Normalized DELs for the blade-root flapwise moment.

The medium- and low-fidelity models show an ability to capture the general trends of the high-fidelity results, reflecting both the M-shape discussed above and an amplitude that is reflective of the high-fidelity results. However, the results are not extremely accurate, sometimes underpredicting or overpredicting the DELs. Moreover, the results from the medium- and low-fidelity models are generally very similar except at the edges of the lateral positions, where the medium-fidelity model tends to be slightly more accurate than the low-fidelity model due to its inclusion of the spatial variation of TI. Thus, there are two important conclusions. First, the use of spatially varying mean wind speed and TI profiles can capture general trends in the blade flapwise DEL and has an adequate degree of accuracy for some of the cases. Second, that the addition of a profile function for the turbulence standard deviation does not add a significant level of accuracy for blade flapwise DELs.

### 3.3.2. Blade Root Edgewise Moment

The normalized DELs for the blade-root edgewise moment are plotted in Figure 9. Because the blade edgewise DEL is dominated by gravitational loads but is still subject to aerodynamic loads, having a wake impinging on one half of the rotor either increases or decreases the DEL depending on whether the wake is located on the left or right side

of the rotor. This phenomena is what causes the characteristic "S"-shape of these curves: when the wake is on the right (left) side of the rotor, then the edgewise DELs are decreased (increased) compared to the freestream DELs. For this load channel, the medium- and low-fidelity models do an excellent job of capturing the DEL. Once again, there is very little difference between the medium- and low-fidelities. Thus, we can conclude that, for the blade-root edgewise DELs, the low-fidelity model is sufficient to adequately calculate the loads in a waked situation.

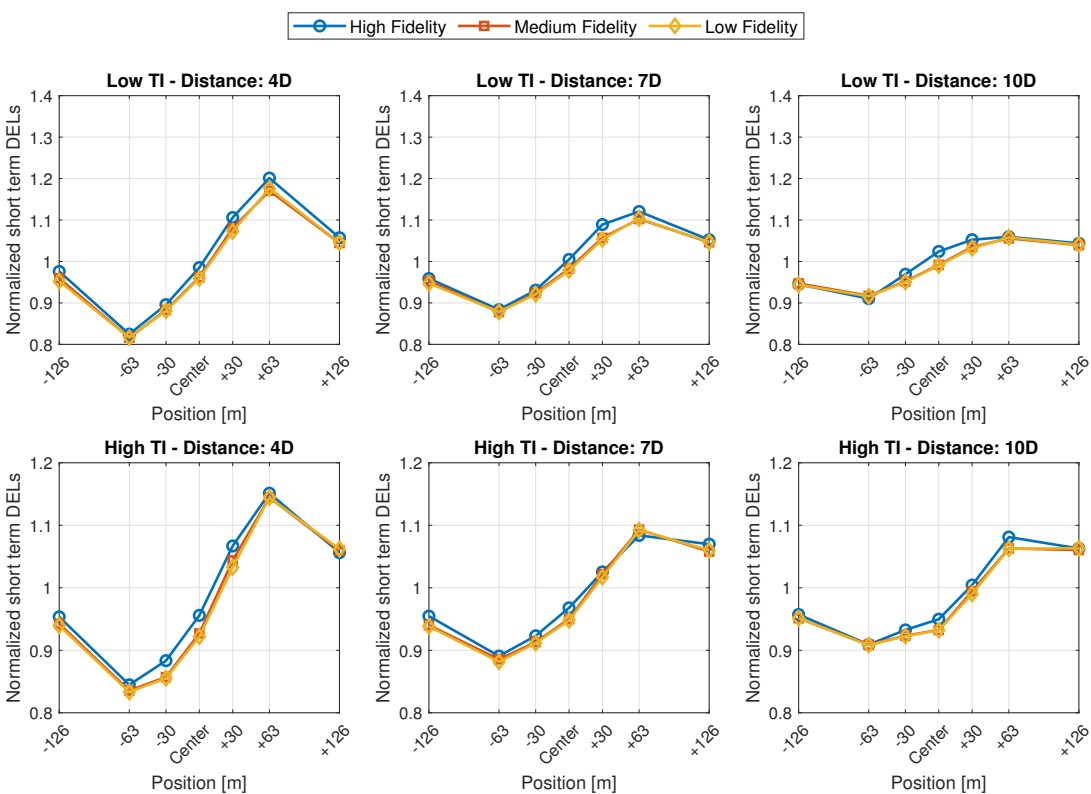

**Figure 9.** Normalized DELs for the blade-root edgewise moment.

### 3.3.3. Shaft Torsion Moment

We now consider the shaft torsional moment. Because the shaft torsion, tower-top roll moment, and tower-bottom side-side moment are closely related load channels, it can be assumed that those two load channels will display similar trends as the shaft torsion.

The DELs for the shaft torsional moment are given in Figure 10. In general, the high-fidelity shaft torsion is shown to be very sensitive to wake conditions. For the low-TI case, the shaft torsion exhibits a similar M-shape as seen in the blade flapwise moment DELs, but the high-TI case also has large DELs near the edges. The shaft torsion's sensitivity to the wake condition is likely caused by the fact that drivetrain modes typically feature very low aeroelastic damping, which more easily propagates unsteady loading into the DELs.

The medium- and low-fidelity models do not perform as well for the shaft torsion as was seen for the previous load channels. Although they capture the M-shaped trend of the low-TI case, there is an offset in the DELs, especially in the $4D$ and $7D$ downstream distances. For the high-TI case, the two lower-fidelity models do not capture the general trend at all, completely missing the reduction in DEL at the center position. The discrepancies decrease for larger downstream distances, but the results are still not extremely accurate. We can therefore conclude that the lower-fidelity models investigated in this paper are generally not accurate for estimating shaft torsion fatigue loads in waked scenarios. Further discussion on this topic is presented in Section 4.

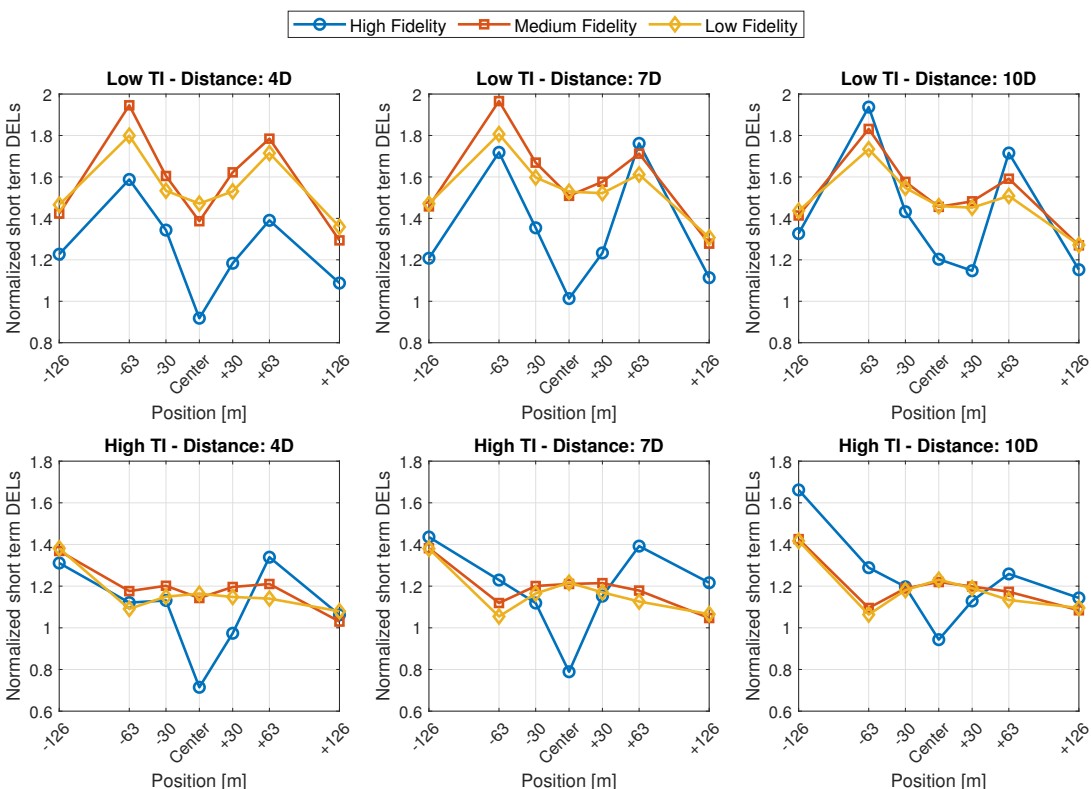

**Figure 10.** Normalized DELs for shaft torsional moment.

### 3.3.4. Tower Bottom Fore-Aft Moment

The final load channel of interest is the tower-bottom fore-aft, which reflects changes in the aerodynamic thrust and rotor deflections and should feature similar characteristics to the tower-top tilt moment (not shown for brevity).

The DELs for the tower-bottom fore-aft are shown in Figure 11. Unlike the previous load channels, this load channel is parabolic-shaped, with the highest DELs near the center. This is expected, as the motion of the wake into and out of the rotor will cause large changes in thrust, resulting in large changes in the tower-bottom load response that will increase the DELs. The height of the parabola decreases with increasing downstream distance, as the wake is not only recovered more at those distances, but also wanders over a wider lateral distance. In general, the lower-fidelity models underpredict the DELs, as they do not feature wake meandering. However, there is finally a significant improvement in the medium-fidelity results over the low-fidelity results, although there are still substantial discrepancies between the medium- and high-fidelity results. Thus, although the steady-state medium- and low-fidelity models can capture the general trends of the tower-bottom fore-aft DEL of a downstream turbine, an accurate calculation requires inflow with wake meandering.

### 3.4. Summary of Results

The medium- and low-fidelity models—steady-state wake models that prescribe profile functions for the mean wind speed and turbulence standard deviation in the YZ plane—are more accurate for certain calculations than for others. The spatial variation of the longitudinal TI for the low-TI, medium-fidelity case generally matched the LES data in the freestream case (see Figure 4), but there was significantly more variation in the downstream cases, in the high-TI case, and in the low-fidelity model. When the medium- and low-fidelity models were used to calculate the mean value of the rotor speed or generator power as a function of downstream/lateral distance, they were both quite accurate for all downstream and lateral positions and for both TI cases. The DELs, however, were

much more varied. The blade-root flapwise DEL calculated with the lower-fidelity models captured the general trends of the high-fidelity DELs with middling accuracy, showing a maximum difference of about 20% for a single lateral location at the 4*D* downstream location. The blade root edgewise DEL had even better prediction ability, capturing the high-fidelity trends quite accurately with a maximum difference of about 4%. The shaft torsion DELs, however, were not adequately captured by the medium- or low-fidelity models, showing discrepancies as large as 60% for some cases. Moreover, the accuracy of the medium-fidelity DELs was not significantly better than the low-fidelity DEL accuracy. This was not the case for the tower-bottom fore-aft DELs, which also had significant discrepancies (up to 33%) between the high-fidelity DELs and the lower-fidelity DELs, but had improved accuracy for the medium-fidelity model. Thus, there is significant variation in the accuracy of the medium- and low-fidelity models depending on the quantity of interest, and it is essential to know this accuracy when attempting to use lower-fidelity models of this kind to calculate operational parameters in turbines. Further discussion on this topic is presented in the section below.

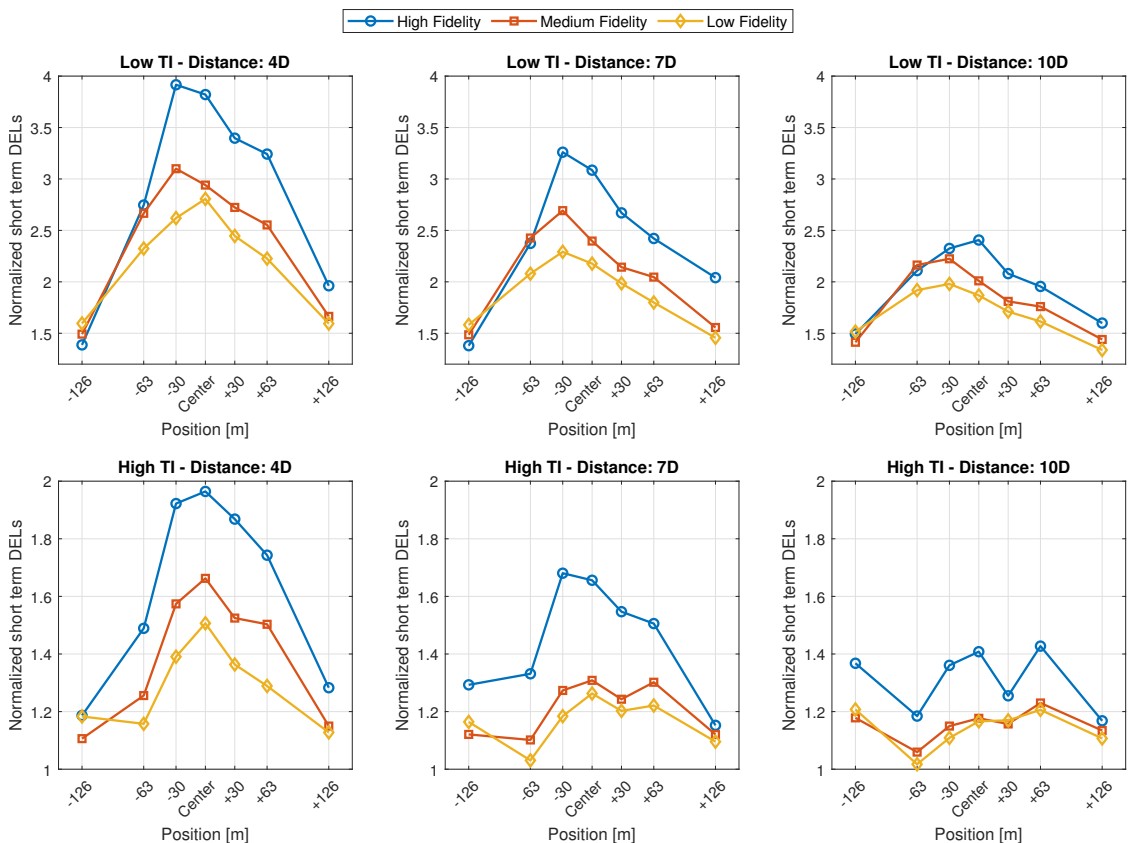

**Figure 11.** Normalized DELs for the tower-bottom fore-aft moment.

## 4. Discussion and Future Work

It should be emphasized that the purpose of this work is not to claim that certain wind turbine parameters such as DELs can be accurately calculated using the two lower-fidelity models presented here when in a waked situation. On the contrary, the presented results clearly indicate that is not the case. Instead, the central objective of this paper is to highlight which quantities of interest, if any, can be predicted with lower-fidelity models and which cannot. A secondary objective is to investigate whether the addition of a profile function for the turbulence standard deviation yields benefits on the calculation accuracy for specific quantities of interest. This information can then be used to make more intelligent decisions in applications like wind-farm-layout planning, where it may be desirable for simple wake

models to be used to estimate the loads of turbines in the farm. Future comparisons with other wake models, both steady-state and dynamic, can yield further insight into the trade-off of computational complexity and accuracy of certain quantities of interest for turbines in waked situation.

One interesting interpretation of the results here is to consider the accuracy in the medium- and low-fidelity models to be a reflection of the impact of wake meandering on that load channel. If we consider the DELs, for example, the blade edgewise loads were very well captured by the lower-fidelity models. This is expected, as edgewise loads are dominated by gravity and only experience small contributions from aerodynamic effects. One the other hand, flapwise fatigue loads are dominated by aerodynamic effects, and yet the blade-root flapwise moment was relatively well captured by the lower-fidelity models. The shaft torsion and tower-base fore-aft moment, however, were not well predicted using the low- and medium-fidelity models. Thus, we can observe that the DELs for the shaft torsion and tower-base fore-aft moment, at least for the simulations investigated here, are more sensitive to meandering than the blade-root loads. This is logical, as the drivetrain mode is very lightly aeroelastically damped and the variation on the thrust force is highly dependent on the location of the wake center, but it should be confirmed with a different turbine model and perhaps a different wake model.

Finally, the results of this paper might not be applicable to wind farms that seek to use wake steering due to the shape of the wake. The LES data that was used in this paper did not include any yaw misalignment with the upstream turbine, which results in a symmetric wake. However, in [27], it was observed that the shape of the wake under yawed conditions is elliptical rather than circular, which has an impact on the estimation of the wake center. The paper further explains that these changes in the wake shape are mainly due to counter-rotating vortices that appear in the flow behind a yawed turbine and generate this distortion. This might be important to consider, in which case LES data for a turbine in a yawed configuration would be needed in future investigations.

## 5. Conclusions

When designing and evaluating wind turbine performance in farms, it can be tempting to calculate loads using simplistic, steady-state models or to neglect loads entirely. However, the impact of modeling loads of waked turbines using steady-state models has not been adequately demonstrated in the literature.

This paper investigates the accuracy of different operational parameters and loads for a downstream turbine when the wake inflow is calculated with different levels of fidelity. The high-fidelity model is extracted directly from two sets of LES data, one with low TI and one with high TI, in which an NREL 5 MW turbine is operating at a fixed pitch angle and RPM. Multiple turbulence boxes are extracted from the LES data at different downstream and lateral locations, as well as a slice $2D$ upstream, which is used as a baseline for the freestream condition. From the high-fidelity turbulence boxes, profile functions that prescribe the variation of the mean wind speed and the turbulence standard deviation as a function of location in the YZ plane are defined. Both profile functions were used to create medium-fidelity turbulence boxes, where the turbulent variations were generated using a Kaimal spectrum with exponential coherence. Low-fidelity turbulence boxes were also simulated by setting the turbulence standard deviation to a constant over the whole rotor.

The accuracy of the results from the medium- and low-fidelity models depended greatly on the quantity of interest, as well as which TI case and downstream location was under investigation. The mean values for the power and rotor speed matched very well, indicating only a slight dependence on wake meandering. The DELs for the blade edgewise loads matched very well, showing an S-shaped trend in the partial-wake scenario caused by the wake location counter-acting the gravitational load. The fatigue loads for the blade flapwise loads captured the correct trends, but featured some substantial differences at certain downstream/lateral locations. The shaft torsion and tower-base fore-aft moment showed the largest sensitivity to wake meandering, where the medium- and low-fidelity

models were not able to capture the trends well, even for larger downstream distances where the wake is less pronounced. The addition of a profile function for the turbulence standard deviation did not significantly increase the accuracy of the DELs except for the tower-base fore-aft moment. This paper concludes that profile functions can be used to estimate blade loads in wind farms, but caution should be taken when applying to loads related to the shaft torsion or to the tower-base fore-aft bending moment. The non-stationarity of wake meandering causes impacts in the DELs that are not captured by the simple models presented here.

For more details on some of the results presented in this paper, please see [17].

**Author Contributions:** The majority of research efforts were completed by E.S.S. The paper was written by J.M.R. with feedback from E.S.S. and L.B. The contributions in detail are as follows: conceptualization, E.S.S., J.M.R. and L.B.; methodology, E.S.S., J.M.R. and L.B.; software, J.M.R. and E.S.S.; validation, E.S.S., J.M.R. and L.B.; formal analysis, E.S.S.; investigation, E.S.S.; resources, J.M.R.; data curation, E.S.S. and J.M.R.; writing—original draft preparation, J.M.R. and E.S.S.; writing—review and editing, E.S.S., J.M.R. and L.B.; visualization, E.S.S.; supervision, J.M.R. and L.B.; project administration, E.S.S., J.M.R. and L.B. All authors have read and agreed to the published version of the manuscript.

**Funding:** This research received no external funding.

**Acknowledgments:** The authors gratefully acknowledge Paul Fleming and Senu Sirnivas from the National Renewable Energy Laboratory for sharing the SOWFA simulations used in this paper.

**Conflicts of Interest:** The authors declare no conflict of interest.

## Appendix A. SOWFA Simulation Parameters

Table A1 contains the parameters used for the SOWFA simulations.

**Table A1.** Parameters used for the set up of the large-eddy simulations in SOWFA for the NREL 5MW Reference Wind Turbine at wind speed 8 m/s. The high-TI case (average TI of 11.0 %) used a surface roughness of 0.15 m; the low-TI case (average TI of 5.6 %), 0.0002 m.

| Domain Size and Number of Cells | |
|---|---|
| Minimum x-extent of domain (m) | 0.0 |
| Minimum y-extent of domain (m) | 0.0 |
| Minimum z-extent of domain (m) | 0.0 |
| Maximum x-extent of domain (m) | 5000.0 |
| Maximum y-extent of domain (m) | 2000.0 |
| Maximum z-extent of domain (m) | 1000.0 |
| Number of cells in x-direction | 500 |
| Number of cells in y-direction | 200 |
| Number of cells in z-direction | 100 |
| **Initial Values for the Variables** | |
| Initial condition for wind speed (m/s) | 8.0 |
| Initial condition for wind direction (deg) | 270.0 |
| Height at which to drive mean wind to U0Mag/dir (m) | 90.0 |
| Initial pressure ($m^2/s^2$) | 0.0 |
| Initial SGS viscosity ($m^2/s$) | 0.0 |
| Initial SGS turbulent kinetic energy ($m^2/s^2$) | 0.1 |
| Initial SGS temperature diffusivity ($m^2/s$) | 0.0 |
| Potential temperature gradient above the strong inversion (K/m) | 0.003 |
| Height of the middle of the initial strong capping inversion (m) | 750.0 |
| Vertical width of the initial strong capping inversion (m) | 100.0 |
| Initial potential temp. at bottom of strong capping inversion (K) | 300.0 |
| Initial potential temp. at top of strong capping inversion (K) | 305 |

**Table A1.** *Cont.*

| General Conditions and Parameters | |
|---|---|
| Molecular Prandtl number | 0.7 |
| Turbulent Prandtl number | 0.33333333 |
| Molecular viscosity ($m^2/s$) | $1.00 \times 10^{-5}$ |
| Reference potential temperature (K) | 300 |
| Latitude on the Earth of the site (deg) | 41.3 |
| Earth's rotation period (hr) | 24 |
| **Surface Conditions** | |
| Temperature flux at wall | (0.0 0.0 0.0) |
| Initial wall shear stress ($m^2/s^2$) | (0.0 0.0 0.0 0.0 0.0 0.0) |
| von Karman constant | 0.4 |
| Surface roughness (m) for High TI case | 0.15 |
| Surface roughness (m) for Low TI case | 0.0002 |
| Treat surface stress wall model | "planarAverage" |
| Monin-Obukhov wall shear stress model constant | 16.0 |
| Monin-Obukhov wall shear stress model constant | 5.0 |
| Monin-Obukhov wall temperature flux model constant | 9.0 |
| Monin-Obukhov wall temperature flux model constant | 7.8 |
| Monin-Obukhov wall temperature flux model constant | 1.0 |

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
