# Peer review of "The Importance of Wake Meandering on Wind Turbine Fatigue Loads in Wake"

_energies, doi:10.3390/en14217313_

Round 1
Reviewer 1 Report
Thank you for a well written and interesting paper. All steps were described in a detailed and careful manner. The discussion is broad and self-critical, yet conclusive. In my opinion no revision is necessary.
Author Response
Thank you kindly for your time in reviewing. We are happy to hear that the effort spent in writing the paper has resulted in a positive review with no changes requested.
Reviewer 2 Report
The paper presents the results of many simulations concerning a steady-state analysis of different parameters of wind turbines especially turbulence parameters.
The presented results are interesting and correct. The paper is written very carefully using precise scientific language.
I have only a few comments for the Authors:
- In my opinion, in the label of figures, some details should be written in the text of the paper, not in the label of figures (Figure 1 and Figure 2).
- There is no information about the colors (some scale, values) in figure 2.
- The time step is fixed and equal to 0.02 seconds. I think there is no sufficient explanation if this value is correctly chosen.
- On figure 5 the curves are not symmetrical. In my opinion, there should be a symmetry of results.
- The results presented in the paper are only numerical. It would be very interesting to try to verify these results with some real experiments (measurements).
- These results are for steady-state. The dynamic behavior of wind turbines is more complicated but more unexplored. Maybe you will try to extend your research for the transient state.
I recommend this paper to be published in Energies.
Author Response
Thank you kindly for your time in reviewing and for your relevant feedback on how to improve the paper. We have addressed your six comments individually:
- We agree that there was room for improvement on the labels of Figures 1 and 2. We have reduced the label captions to only the most relevant information and placed more supporting text in the main text when the figures are introduced.
- Because the contours are meant to be indicative, not quantitative, we chose to omit a colorbar or discussion of the wind speed values to reduce the amount of information being presented to the reader. We would prefer to keep the text as it stands now, to reduce the mental load on the reader.
- Thank you for noticing this omission. The time step of 0.02 seconds is what was chosen by the model developers to ensure model convergence but minimal computation time. We have added some text explaining that this is the standard time step for the model.
- The curves are asymmetric due to the 700-second turbulence box being too short to achieve spatial stationarity. We have added some text to clarify this when Figure 5 is analysed in the main text.
- We agree that it would be interesting to try out this method with experiments. The logistics of such a campaign, unfortunately, are much beyond the scope of this paper.
- These results do include fatigue loads, which are relatively sensitive to transience as we concluded in this paper. However we agree that it would be very interesting to consider in future work considering error metrics such as the mean absolute error between two time signals.
Reviewer 3 Report
Dear authors,
the article describes the effect of a previous wind turbine to the loads on the next one. The study is fully numerical, it interesting and, according to my insight, the methods seem to be relevant and propperly used (however, I am not a CFD expert, therefore I am not able to detect some error in the details). There are two interesting results mixed: first is the physical effect of the layout, which can be really interesting for a engineer planing the farm. The second and very different result, is the study, which simplification level of the numerical method is sufficient for this problem. This is a very interesting for an programmer. I think, that it would be better to somehow better separate the mentioned conclusions, becasue they are interesting for different groups of readers.
- Not all abbreviations are explained: CFD, HAWC2, SOWFA.
- the turbulence intensity TI should be written with subscript in equations (in text it is OK)
- I do not see subscript G in the Figure 1
- D in meaning of rotor diameter should be written in italic (otherwise e.g. 2D means "two-dimensional")
- It would be nice to show the structure of turbulence (e.g. length-scales and anizotropy), and how it impacts to the DEL.
- It would be excellent to compare with some experimental data. E.g. to show the damage on some turbine, which is already operating some time.
- In the conclusions, it should be added some recommendation for the farm design: e.g. "aligning turbines decreases the power output, but on the other hand, it saves the life-time due to smaller loads" or something like this explicitly mentioning, that the fatigue loads are an issue which has be taken into account during the farm design.
I think, that the article is very good and interesting and the mentioned points are not needed, they are just recommendations and I understand, that it is not always possible to have additional experimental data and so on.
Author Response
Thank you kindly for your time in reviewing and for your feedback on how to improve the paper. We have addressed your comments individually:
- Some abbreviations not defined. We have added the missing definitions.
- The turbulence intensity (TI) should be written with a subscript in equations. The turbulence intensity is traditionally indicated by variable I in equations (see, e.g., IEC 61400-1). Thus, we have changed the variable to be consistent with existing literature.
- Subscript G is not seen in Figure 1. That sentence was a remnant from a previous iteration of the figure, and it has therefore been removed.
- Variable D should be written in italic. All occurrences of the variable have been changed to italics in the text.
- It would be nice to show the turbulence structure and impact on the DEL. We agree that a more in-depth discussion on the power spectra and structural properties of the turbulence and the resulting impact on fatigue loads would be very interesting. Unfortunately, as the paper is already quite long we will have to leave these investigations for future work.
- It would be excellent to compare with experimental data. Although an inclusion of experimental data would be extremely interesting, the scope of such work is well beyond the contents of this paper. We hope to have experimental data in the future that we could use in such a study.
- The conclusion section should have an explicit reference to farm design. We have added text to the conclusion explaining the relationship and relevance of this work to farm design.